# Persistence of *Brucella abortus* lineages revealed by genomic characterization and phylodynamic analysis

**Marcela Suárez-Esquivel**[1], **Gabriela Hernández-Mora**[2], **Nazareth Ruiz-Villalobos**[1], **Elías Barquero-Calvo** [1,3], **Carlos Chacón-Díaz**[3], **Jason T. Ladner**[4], **Gerardo Oviedo-Sánchez**[1,3], **Jeffrey T. Foster**[4], **Norman Rojas-Campos**[3], **Esteban Chaves-Olarte**[3], **Nicholas R. Thomson**[5], **Edgardo Moreno**[1], **Caterina Guzmán-Verri**[1,3]*

1 Programa de Investigación en Enfermedades Tropicales, Escuela de Medicina Veterinaria, Universidad Nacional, Heredia, Costa Rica, 2 Servicio Nacional de Salud Animal, Ministerio de Agricultura y Ganadería, Heredia, Costa Rica, 3 Centro de Investigación en Enfermedades Tropicales, Facultad de Microbiología, Universidad de Costa Rica, San José, Costa Rica, 4 The Pathogen and Microbiome Institute, Northern Arizona University, United States of America, 5 Parasites and Microbes from Pathogen Genomics, Wellcome Trust Sanger Institute, Hinxton, United Kingdom

* catguz@una.cr

**Data Availability Statement:** All relevant data are within the manuscript and its Supporting Information files.

## Abstract

Brucellosis, caused by *Brucella abortus*, is a major disease of cattle and humans worldwide distributed. Eradication and control of the disease has been difficult in Central and South America, Central Asia, the Mediterranean and the Middle East. Epidemiological strategies combined with phylogenetic methods provide the high-resolution power needed to study relationships between surveillance data and pathogen population dynamics, using genetic diversity and spatiotemporal distributions. This information is crucial for prevention and control of disease spreading at a local and worldwide level. In Costa Rica (CR), the disease was first reported at the beginning of the 20th century and has not been controlled despite many efforts. We characterized 188 *B. abortus* isolates from CR recovered from cattle, humans and water buffalo, from 2003 to 2018, and whole genome sequencing (WGS) was performed in 95 of them. They were also assessed based on geographic origin, date of introduction, and phylogenetic associations in a worldwide and national context. Our results show circulation of five *B. abortus* lineages (I to V) in CR, phylogenetically related to isolates from the United States, United Kingdom, and South America. Lineage I was dominant and probably introduced at the end of the 19th century. Lineage II, represented by a single isolate from a water buffalo, clustered with a Colombian sample, and was likely introduced after 1845. Lineages III and IV were likely introduced during the early 2000s. Fourteen isolates from humans were found within the same lineage (lineage I) regardless of their geographic origin within the country. The main CR lineages, introduced more than 100 years ago, are widely spread throughout the country, in contrast to new introductions that seemed to be more geographically restricted. Following the brucellosis prevalence and the farming practices of several middle- and low-income countries, similar scenarios could be found in other regions worldwide.

**Funding:** This work was supported by FEES-CONARE, Costa Rica, FIDA-UNA 0045-17, Costa Rica, UCREA- Universidad de Costa Rica (Grant B8762) and Wellcome Trust. MS-E and NRV were partially sponsored by UCR scholarships. NRT was supported by Wellcome Trust [098051]. JT Ladner and JT Foster were funded under the State of Arizona Technology and Research Initiative Fund (TRIF), administered by the Arizona Board of Regents, through Northern Arizona University. The funders had no role in study design, data collection and analysis, decision to publish, or preparation of the manuscript.

**Competing interests:** The authors have declared that no competing interests exist.

## Author summary

Brucellosis caused by *Brucella abortus* is an important disease of cattle and humans, highly prevalent in middle- and low-income countries worldwide. Determining the prevalent *B. abortus* lineages as well as the sources and timing of strain introductions into a country, is necessary for understanding the epidemiology and the natural history of the disease. Our results, derived from 188 *B. abortus* isolates from Costa Rica and 221 genomes from the same species worldwide, showed that in spite of being a small country (~51,100 km$^2$), there are at least five *B. abortus* lineages circulating in its territory. Furthermore, regardless of control measures, the lineages introduced more than 100 years ago are still present and widespread throughout the country, in contrast to more recent introductions that seemed to be geographically restricted. By means of our analysis, we constructed a road map based on an integrative approach that may improve the understanding of the disease dynamics. Following the brucellosis prevalence and the farming practices of several middle- and low-income countries, similar scenarios could be found in other regions worldwide, where our methods could be useful and similarly applied.

## Introduction

Members of the genus *Brucella* are Gram-negative facultative extracellular intracellular bacteria that infect a variety of animals, including humans [1,2]. In domestic livestock such as cows, goats, sheep and pigs, *Brucella* species induce abortion and orchiepididymitis, causing significant economic losses, mainly in middle- and low-income countries. Human brucellosis is a chronic debilitating disease and if not treated, may cause death. The exact number of worldwide animal and human brucellosis cases is not known, but it is projected high [3]. Just in Inner Mongolia, China, the incidence of human brucellosis was estimated to be close to 300,000 new cases from 2010–2014 [4]. In spite of this, according to WHO Resolution WHA66.12 from 2014, brucellosis was not included as a zoonotic neglected disease and was classified as a "tool-deficient" disease for which better control methods need to be developed [5,6].

Costa Rica is a tropical and subtropical Central American country with a land area of ~51,100 km$^2$. Socioeconomically, the country is divided in six regions: Northern, Central, Brunca, Chorotega, Caribbean Huetar and Central Pacific [7]; the Northern and Central regions are the main livestock farming areas. The only prevalent *Brucella* species in domestic livestock of CR is *Brucella abortus*, with a seroprevalence in cattle close to 11% and to 21.7% in water buffalo farms [8,9]. No information of brucellosis in CR terrestrial wildlife is available. Brucellosis prevalence in cattle has been reported as 0.5–10% in Latin American countries, and from 4–11% in Central American countries [10–12].

In contrast to other American countries, the introduction of European cattle to CR did not occur with the arrival of explorers from Europe. The first record of cattle importation into the CR territory dates to 1561 [13] and subsequently different cattle breeds arrived to CR from 1568–1920, as it is recorded in several documents. Those documents also report recurrent abortions in the Central Valley and in the highlands, suggesting the occurrence of brucellosis [13]. On the other hand, records indicate that the introduction of water buffalo occurred in 1974 [14]. The first *B. abortus* isolates recovered in Costa Rica from buffalo are from 2018, reported in this study.

Brucellosis became a notifiable disease in CR in 1915, after the first isolation of "Bang´s bacillus" from the blood of a human patient [15,16]. More recently, and thanks to improved surveillance and diagnosis of the disease by public health authorities [8], an increasing number of human cases have been reported, with positive hemocultures and description of the first human cases caused by *Brucella neotomae* [17,18].

In a previous study, focused on prevalence and molecular epidemiology of *B. abortus* in bovines from CR, we showed that the isolates clustered in four discrete groups using Multiple-Locus Variable Number Tandem Repeat Analysis of 16 loci (MLVA-16). This suggested at least four different introductions of the bacterium into CR [8]. To gain insights on the phylodynamics of *B. abortus* introductions, we characterized a total of 188 *B. abortus* isolates from CR recovered from bovines and humans from 2003 to 2018, by MLVA and 95 of them by whole genome sequencing (WGS). Then, we explored the origin and dates of the introduction of circulating *B. abortus* strains in CR to assess the timing and spread of this disease. We found five distinct *B. abortus* lineages, and their phylodynamics associated their most recent common ancestor to known introduction dates of bovine species and breeds into the country.

Here, we propose a model on how to study the spread of brucellosis in a particular time frame and geographic context. Our model can be reproduced in other regions were brucellosis is endemic to track and assess factors associated with the spread and maintenance of this disease.

## Methods

### Ethics statement

The genetic resources were accessed in Costa Rica according to the Biodiversity Law #7788 and the Convention on Biological Diversity, under the terms of respect to equal and fair distribution of benefits among those who provided such resources under CONAGEBIO Costa Rica permits # R-028-203-OT and # R-CM-UNA-003-2019-OT-CONAGEBIO. All data analyzed were anonymized.

### *Brucella* isolates

A total of 188 *B. abortus* clinical isolates recovered in CR during 2003 to 2018 were obtained through the brucellosis national surveillance programs, from Rose Bengal serology test positive animals or humans. Cattle isolates comprised the majority of samples (n = 162), followed by human (n = 16) and water buffalo (n = 10) (Table 1). Of those, 54 isolates from bovines from the province of Cartago were recovered during an outbreak that occurred from 2003 to 2005 in an 18.7 km$^2$ and geographically restricted area named San Juan de Chicuá. The isolates were characterized by phenotypic and molecular methods as previously described [8]. Briefly, cultures were performed in non-selective and selective media including blood agar and Columbia agar, supplemented with 5% dextrose and sheep blood. Cultures were incubated in 10% $CO_2$

**Table 1. Hosts and geographic origin of *Brucella abortus* isolates of Costa Rica.** The Northern and Central regions are the main CR dairy and beef producers.

| Host | North | Central | Brunca | Chorotega | Caribbean Huetar | Central Pacific | ND[a] | Total |
|------|-------|---------|--------|-----------|------------------|-----------------|------|-------|
| Cattle | 28 | 110 | 1 | 6 | 13 | 1 | 3 | 162 |
| Human | 4 | 9 | 0 | 1 | 2 | 0 | 0 | 16 |
| Water Buffalo | 6 | 0 | 0 | 1 | 3 | 0 | 0 | 10 |
| **Total** | 38 | 119 | 1 | 8 | 18 | 1 | 3 | 188 |

[a]ND: no data

atmosphere at 37˚C for at least two weeks. Bacterial colonies similar to *Brucella* sp were subjected to Gram staining, agglutination with acriflavine and acridine orange dyes and tested for urease and oxidase activity, citrate utilization, nitrate reduction, $H_2S$ production, growth in the presence of thionin (20 μg/mL) and basic fuchsin (20 μg/mL) and uptake of crystal violet. All procedures involving live *Brucella* were carried out according to the "Reglamento de Bioseguridad de la CCSS 39975–0", year 2012, after the "Decreto Ejecutivo #30965-S", year 2002. WGS was performed for 95 isolates, including all human isolates and the rest selected according to MLVA-16 clustering and geographical distribution (S1 Dataset).

In order to gain phylogeographic context for the MLVA-16 and WGS analysis, data from *B. abortus* isolates from other countries, and available in the public databases (http://microbesgenotyping.i2bc.paris-saclay.fr; https://www.ncbi.nlm.nih.gov/genome/genomes/) were also included (S1 Dataset).

## MLVA and whole genome sequencing

DNA was extracted with a DNeasy Blood & Tissue kit from QIAGEN or Promega Wizard Genomic DNA Purification kit, and stored at -70˚C until use. Bruce-ladder multiplex PCR analysis was performed as previously described [19]. MLVA-16 and the corresponding cladograms were carried out as reported (http://mlva.i2bc.paris-saclay.fr/brucella/spip.php?rubrique29) including 502 isolates worldwide [8]. Of the 188 isolates from CR, 149 were retained in the dendrogram as 39 samples had identical MLVA-16 profiles and were excluded to increase resolution. Values obtained for each MLVA marker are in S1 Dataset.

WGS was performed both at the Wellcome Trust Sanger Institute and Centro de Investigación en Biología Celular y Molecular (CIBCM) of Universidad de CR (UCR) on Illumina platforms according to in house protocols [20,21]. For genome assembly, sequencing reads were *de novo* assembled using Velvet Optimiser [22] and contigs were ordered by Abacas [23], using *B. abortus* 9–941 (NCBI accession numbers NC_006932 and NC_006933) as the reference. To detect mis-assemblies, raw data were mapped back to the *de novo* genome assemblies using SMALT v.0.5.8 (http://www.sanger.ac.uk/resources/software/smalt/). All sequencing data have been deposited at the European Nucleotide Archive (ENA) (http://www.ebi.ac.uk/ena/) under the accession codes listed in S1 Dataset. Other WGS from various *Brucella* strains used for comparative purposes were obtained from the NCBI Genome database (S1 Dataset).

A *B. abortus* genome isolated from a human (babohCR175) was sequenced with Oxford Nanopore MinION technology [24], and the assembly was performed along with Illumina reads by Unicycler [25].

## Phylogenetic reconstruction and WGS based analysis

A total of 219 *B. abortus* genomes were used in this analysis; 101 of those generated during this study. The detailed information and metadata of the genomes is presented in S1 Dataset and includes 95 CR isolates: 76 from cattle, 3 from water buffalo, and 16 from humans. These were analyzed alongside 8 genomes from reference strains from other *Brucella* species (*B. canis*, *B. ceti*, *B. melitensis*, *B. microti*, *B. neotomae*, *B. ovis*, *B. pinnipedialis* and *B. suis*) and two *Ochrobactrum* species were used as an outgroup.

To construct a multiple sequence alignment for phylogenetic reconstruction, reads from two *Ochrobactrum* species and the *Brucella* isolates from different hosts (S1 Dataset) were aligned by bwa and mapped with SMALT v.0.5.8 against *B. abortus* 9–941, with an average coverage of 98.75%. Single Nucleotide Polymorphisms (SNPs) were called using Samtools [26], and 322,266 variable sites were extracted using snp sites [27]. The resulting alignment was used for maximum likelihood phylogenetic reconstruction with RAxML v8 [28]. The

phylogenetic tree was rooted using *Ochrobactrum anthropi* ATCC49188 and *O. intermedium* strain LMG3301.

Figtree v1.4.3 (http://tree.bio.ed.ac.uk/software/figtree/) and ggtree [29,30] were used for visualization of the phylogenetic tree, and its relationship with the metadata was facilitated by microreact [31] (https://microreact.org/project/BJJ4H3H-E).

The population structure was inferred by RhierBAPS [32] using the 322,266 core SNPs (S2 Dataset). The analysis was performed with four depth levels and a maximum clustering population size of 45 (default = number of isolates/5; 221/5 = 44.2).

All analyses relevant to reference annotation (e.g. dN/dS calculations and SNP positions in coding sequences) were relative to *B. abortus* 9–941 (accession numbers NC_006932 and NC_006933), as detailed in S3 Dataset.

The putative cellular localization of the coding sequences (CDS) including non-synonymous SNPs, or pseudogenes, was predicted by PSORT and the function was classified based on the product description in the references and the related metabolic pathway according to KEGG and BioCyc [33,34].

**Anomalous regions and repetitive elements analyses.** The presence, orientation and distribution of 23 previously reported genomic islands (GI) or anomalous regions (regions likely acquired by horizontal gene transfer) [35–37] were examined across seven phylogenetically representative *B. abortus* genomes from CR. For this, a "genomic-island pseudo-molecule" was formed by concatenation of 23 genomic regions obtained from several *Brucella* reference sequences, as previously described [38]. A BLAST comparison between the representative genomes and the pseudo-molecule was performed and visualized using ACT (S1 Fig).

The number and position of the insertion sequence IS*711* were searched in the analyzed genomes by mapping the reads with bwa and SMALT v.0.5.8 to the 842 bp IS*711* of *B. ovis* (accession number M94960). Those reads that showed 99% mapping identity to IS*711*, were then mapped again to the full WGS of *B. ovis* ATCC 25480 in order to judge potential insertion sites. The reads that mapped with identity equal to or higher than 90% to the reference were filtered to 50x coverage and used to produce a visual representation displaying the identified sites per genome and approximate location according to *B. ovis* sequence coordinates. The same procedure was followed with the 6002 bp Tn*2020* (accession number AF118548.1), and *B. abortus* 9–941 was used as reference for mapping back the reads that show 99% identity to Tn*2020* sequence (S2 Fig).

**Bayesian time-structured coalescent analysis and tree calibration.** For the molecular clock estimation, Bayesian Evolutionary Analysis Sampling Trees (BEAST) v1.10.1 [39] was used to generate a time-structured phylogeny including only the 110 genomes from the dataset, with known isolation year (S1 Dataset). Any trace of ancestral recombination was removed by ClonalFrameML [40] from the resulting alignment.

The alignments included only variable positions, but the BEAST XML input file was modified to specify the number of invariant sites, by nucleotide, in the *B. abortus* genomes. Six different combinations of molecular clock and coalescent models were evaluated (S2 Dataset) using path-sampling and stepping-stone marginal likelihood estimation approaches [41–43]. Each model combination was run in duplicate, with 1 billion Markov Chain Monte Carlo steps, sampling parameters and trees every 100,000 generations to ensure independent convergence of the chains. The log files were combined with LogCombiner v1.10.1 and assessed with Tracer v1.7.1. The first 100,000,000 iterations were discarded as burn-in.

The BEAST maximum clade credibility (MCC) tree was generated by TreeAnnotator v.1.10.1 and visualized using Figtree v1.4.3 (http://tree.bio.ed.ac.uk/software/figtree/). Baltic (https://github.com/evogytis/baltic) was used for parsing and visualizing results from BEAST trees (code available at https://github.com/msuareze/Manuscripts.git). For CR lineage specific

analyses, data were parsed with samogitia (https://github.com/evogytis/baltic/blob/master/baltic/samogitia.py) from the trees generated from the BEAST analysis with the full data set.

The results of the approximate node ages were compared to historical records kept by the Corporation for Cattle Ranching Enhancement (CORFOGA, Spanish acronym) and from the scientific literature.

## Results

### *B. abortus* phylogeny shows the presence of five lineages in CR

A SNP matrix, including 322,266 sites from 219 *B. abortus* genomes, 8 from other *Brucella* species and two *Ochrobactrum* genomes as an outgroup, was generated and used to produce a maximum likelihood phylogenetic tree, with *B. abortus* 9–941 as a reference. Within this dataset, the *Ochrobactrum* spp. outgroup was separated from the *Brucella* clade by 286,151 SNPs. When limited to only *B. abortus* isolates, 14,736 SNPs were found among the 221 genomes used for this study. The phylogenetic reconstruction showed clustering patterns consistent with the country of origin of the isolates (https://microreact.org/project/BJJ4H3H-E). When looking at the isolates found in Costa Rica (Fig 1) and according to hierBAPS (S2 Dataset) and BEAST, there are at least five distinct lineages currently circulating in the country.

CR lineage I (Fig 1, https://microreact.org/project/BJJ4H3H-E) formed a well-supported CR clade by both bootstrap and divergence level. This lineage containing 83 genomes coming from all the socioeconomic regions of CR (Fig 2), included one genome isolated from buffalo and was related to isolates from the United States (US) and United Kingdom (UK). Moreover, 54 isolates within this lineage were recovered during an outbreak in cattle that occurred from 2003–2005 in San Juan de Chicuá, Central Region and were separated by 1–18 SNPs (S3 Dataset). Interestingly, of those 83 isolates, fourteen obtained from humans were found within this lineage. Of those, nine coming from different country regions are closely related and separated by only 27 SNPs positions from the other genomes included in lineage I (S3 Dataset). We did not detect a connection between these alternative alleles in each SNP position and metabolic pathways. To avoid the loss of any information contained at contig breaks, and/or differences at indel level from human isolates, we confirmed our findings by also sequencing a representative human isolate (babohCR175) using minION Oxford Nanopore (S1 Dataset).

Lineage II was represented by a single isolate from 2018 from a water buffalo and relates to isolates from Colombia, the US, and Mexico. Clade III is well-defined and was composed exclusively of eight CR isolates, 6 obtained from cattle, one from buffalo and one from human; this lineage shares a common ancestor with isolates from the US and UK. Another two CR genomes from 2010 formed Clade IV, with the most closely related genomes coming from the US. Clade V is represented by a single isolate from a human, and is related to genomes from the US; the isolation date is uncertain, but it was likely recovered during 2010.

These five lineages did not correspond to the topology of the MLVA-16 analysis. It has been previously reported [8] that the CR isolates clustered together with samples from South America, US, and Europe, and were distributed among four main clusters. Increasing the number of CR isolates did not change this topology (Fig 3, https://microreact.org/project/ci2RSdpfd).

### Genomic characterization of *B. abortus* lineages reveals variation in genomic islands distribution

We wondered if the *B. abortus* lineages comprised more variation than that revealed by the SNP patterns used for the phylogenetic reconstruction. To asses this question, we looked at known genomic traits associated with variability in *Brucella* [37,38,44,45] that might explain

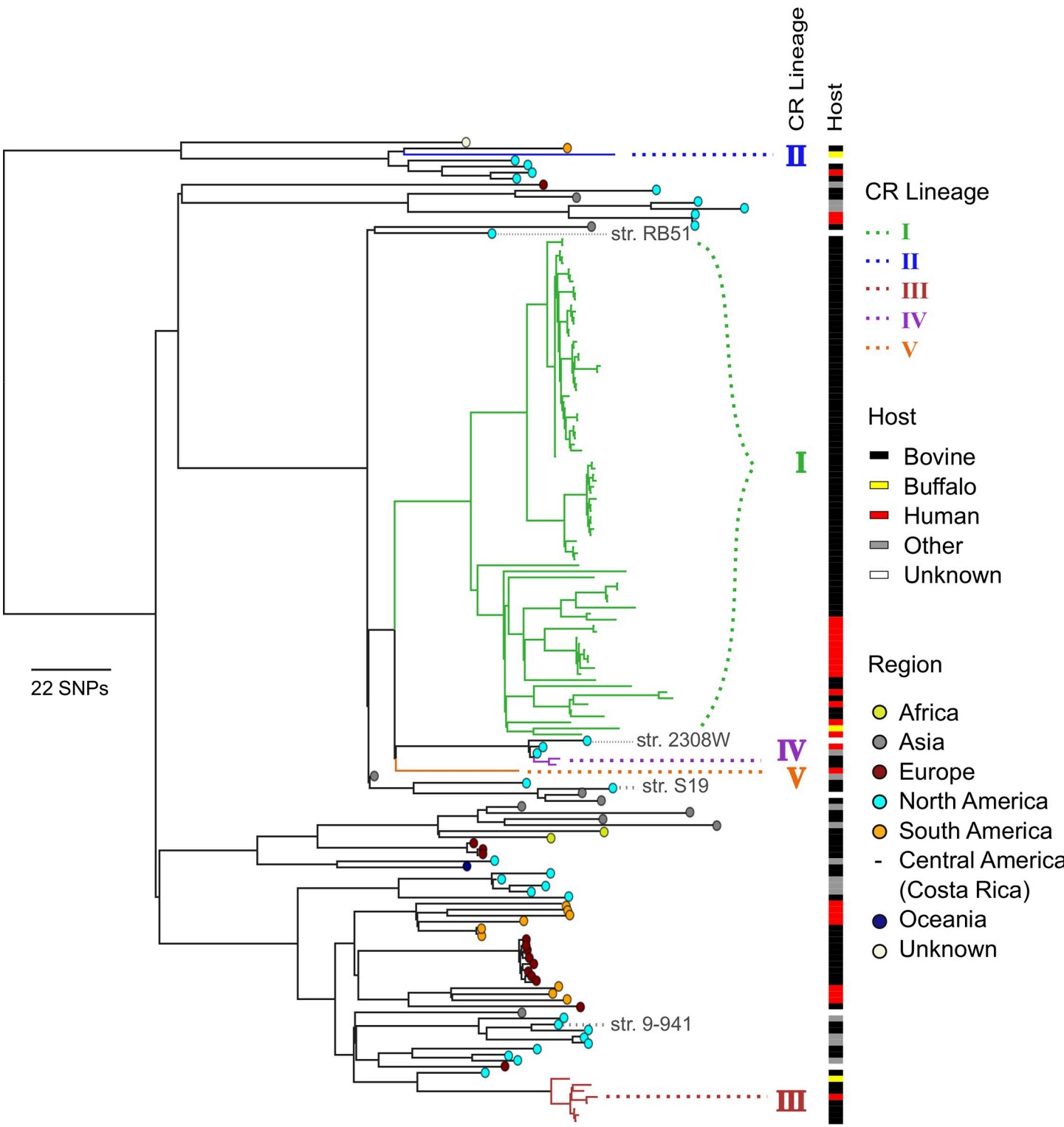

**Fig 1. Phylogeny of *B. abortus* reveals five circulating lineages in CR.** Zoomed representation of clades including the CR lineages from a phylogenetic tree based on 322,266 SNPs of different *Brucella* WGS. *Ochrobactrum*, used as the original root for the tree, species different of *B. abortus*, and several *B. abortus* clades were trimmed from the figure to increase tree resolution of CR lineages. Tip colors indicate the geographic origin of the isolates; colored branches indicate the CR lineages. Colored bars next to tree tips represent the host from which the isolates were recovered. Each clade defining branch showed a 100 bootstrap support value. Several recognized strains are indicated in the tree in gray font to facilitate the visualization of phylogenetic relationships. See https://microreact.org/project/BJJ4H3H-E for further details and full tree.

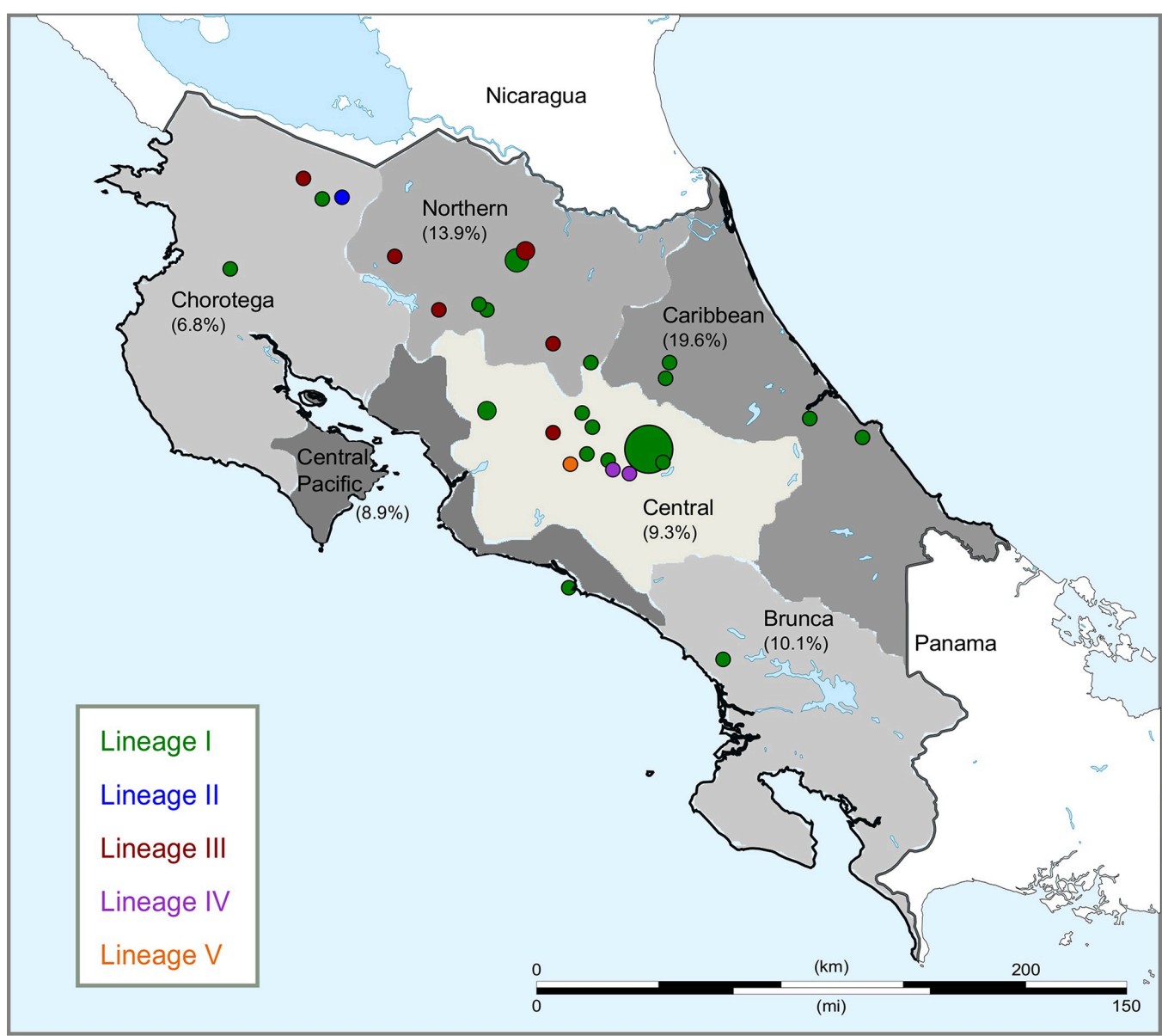

**Fig 2. Map of CR indicating the regions from which *B. abortus* was isolated (circles).** Circle sizes are proportional to the number of isolates recovered from each geographical point, and the colors correspond to lineages I-V. The herd prevalence determined by Rose Bengal test for bovine brucellosis in each socioeconomical region is shown in parenthesis [8]. Prevalence in water buffalo is estimated in 21.7% [9]. Lineage I and III are dispersed along the territory, but are mainly present in the North-East region of CR, where livestock expansion predominates. Map modified from Wikimedia Commons (https://commons.wikimedia.org/wiki/Atlas_of_the_world), Creative Commons CC0 License.

phenotypic or pathogenic differences among the lineages. *De novo* assemblies of representative genomes from each lineage were produced to identify major changes in 23 GIs or anomalous regions. To assess the number and position of the insertion element IS*711* (842 bp) and of the transposon Tn*2020* (6002 bp) [46] within the lineages, reads were re-mapped to a closed genome.

The comparison of GI and anomalous regions among lineages in CR revealed no major deletions or insertions in the *B. abortus* genomes. However, a particular reordering pattern for each one was detected (S1 Fig). This result is relevant since the GIs include transposase and

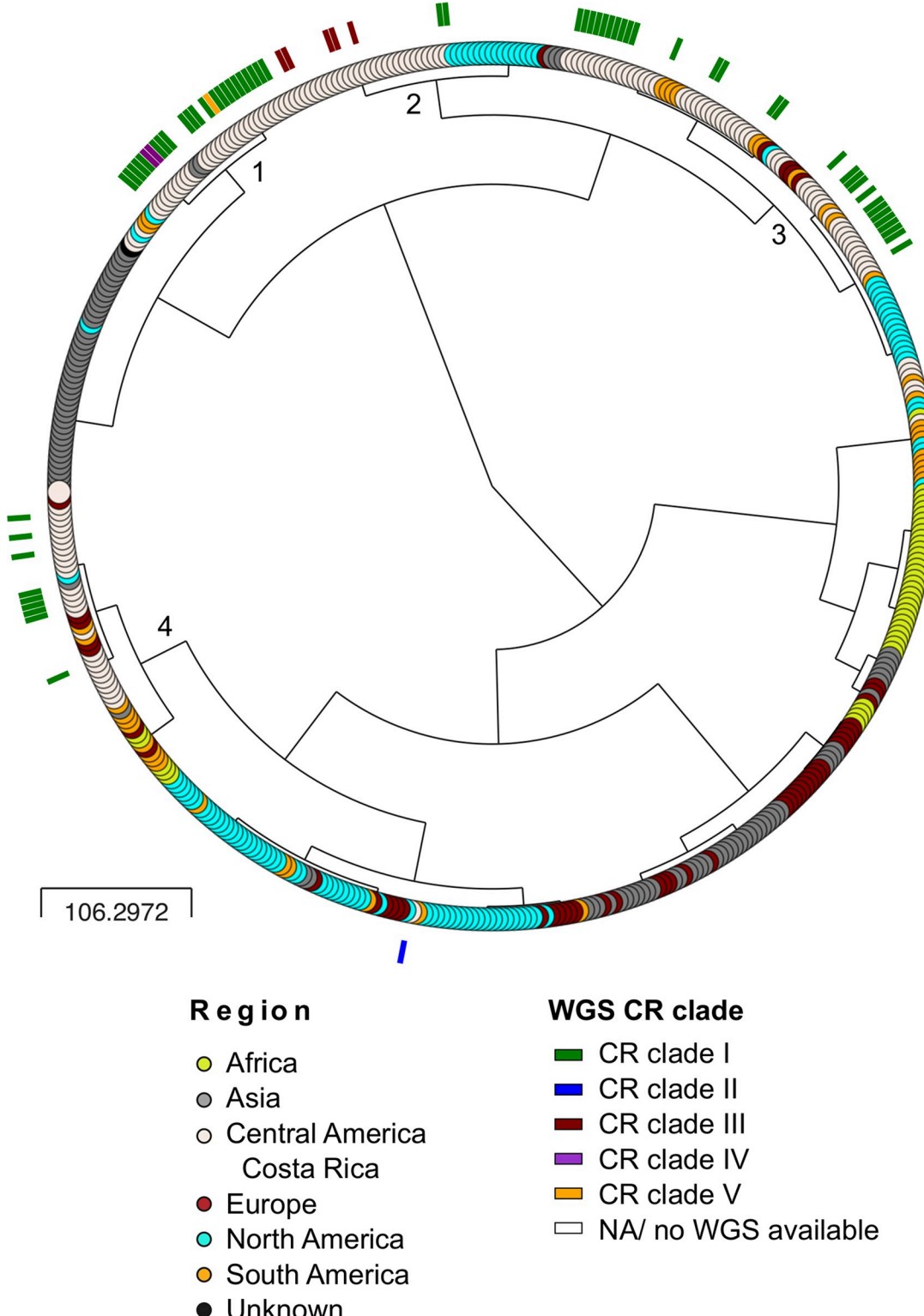

**Fig 3. MLVA-16 cladogram shows four *B. abortus* main clusters.** The analysis of 463 isolates from different countries was performed according to: http://microbesgenotyping.i2bc.paris-saclay.fr/. Tip colors indicate the geographic origin of the isolates and color bars next to the tips represent the corresponding CR lineages found by WGS phylogenetic analysis. The four main MLVA clusters are indicated by an Arabic number next to the branching point. For increased resolution and an interactive view, see https://microreact.org/project/ci2RSdpfd.

transcriptional regulators, among other genes, therefore different positions of these GIs within the chromosome can induce different gene expression patterns.

The number of IS*711* consistently ranged from four to six within the CR lineages. Some were truncated and may indicate genome reduction in comparison to other *B. abortus* genomes [47,48]. Additionally, we found a single copy of Tn*2020* in the same position in the representative genomes (S2 Fig).

### Time-structured analysis of *B. abortus* clusters reveals four introduction events

We used BEAST to determine the timing of the introduction events of four CR *B. abortus* lineages. The isolate of lineage V was excluded from the analysis, as the year of isolation was unknown. The uncorrelated relaxed clock model with the skyline tree was best supported by maximum likelihood analysis (S2 Dataset). Using this model, the mean substitution rate estimated for *B. abortus* was $8.28 \times 10^{-8}$ (95% HPD interval: $2.8 \times 10^{-8}$–$1.7 \times 10^{-7}$) per site per year.

The MCC tree (Fig 4) revealed that at least four introductions of *B. abortus* occurred at different times into CR (Table 2). The lineage I introduction to CR likely occurred around 1899 (95% HPD interval: 1845–1944), close to the first report of bovine brucellosis in the country, from 1897 to 1902 (Figs 4 and 5), in agreement with the importation of dairy breeds from Europe and the US [13].

Lineage II from CR was represented by a single isolate from a water buffalo of the northeast region of the country, recovered in 2018 (Fig 4) and clustered with a 1997 Colombian isolate. This linage showed only an upper bound most recent common ancestor (MRCA), different from other lineages for which both, upper and lower bound, were estimated. Because of this, an approximate time for introduction of lineage II into CR could not be assessed, since the event may have occurred anytime along its branch, likely sometime around 1845 (95% HPD: 1763.2–1905.8). The introduction of water buffalo from India to South America and Trinidad and Tobago (T&T) occurred at the beginning of the 1900s [49,50] and so, considering the MRCA estimation, it is likely that its MRCA diverged in T&T prior to the introduction of water buffalo into CR in 1974 [14] (Fig 5).

Lineages III and IV were more recent introductions. Lineage III MRCA likely occurred sometime between 1907 and 1990, at the time that most creole cattle were replaced with European and South American breeds [51]. Lineage IV MRCA originated between 1978 and 2001. During these periods, records indicate illegal importation of cattle and bovine semen from Nicaragua [51]. Unfortunately, we do not have WGS or isolates from Nicaragua to test this association. Around year 2000 additional water buffalo arrived from Guatemala that serologically tested positive for brucellosis based on the Rose Bengal test. This is consistent with the arrival of a new lineage of *B. abortus* into CR at that time.

## Discussion

In low to middle income countries such as CR, brucellosis is a persistent disease in domestic animals that endures due to local farming practices and the easy movement of animals across borders [52]. This movement of animals or their products, allows the dispersion of infectious agents such as *Brucella* spp. and the introduction of new lineages into those territories. The evidence of foreign lineages of some pathogens in a country, like *Brucella*, allows some understanding of the natural history of the pathogen, but can also reveal inconsistencies in control measures and biosecurity at national borders. We used genomic characterization and phylodynamic analysis of *B. abortus* isolates as a model to understand the evolution of brucellosis in CR, in order to detect the origins and spread of circulating strains. MLVA has been used

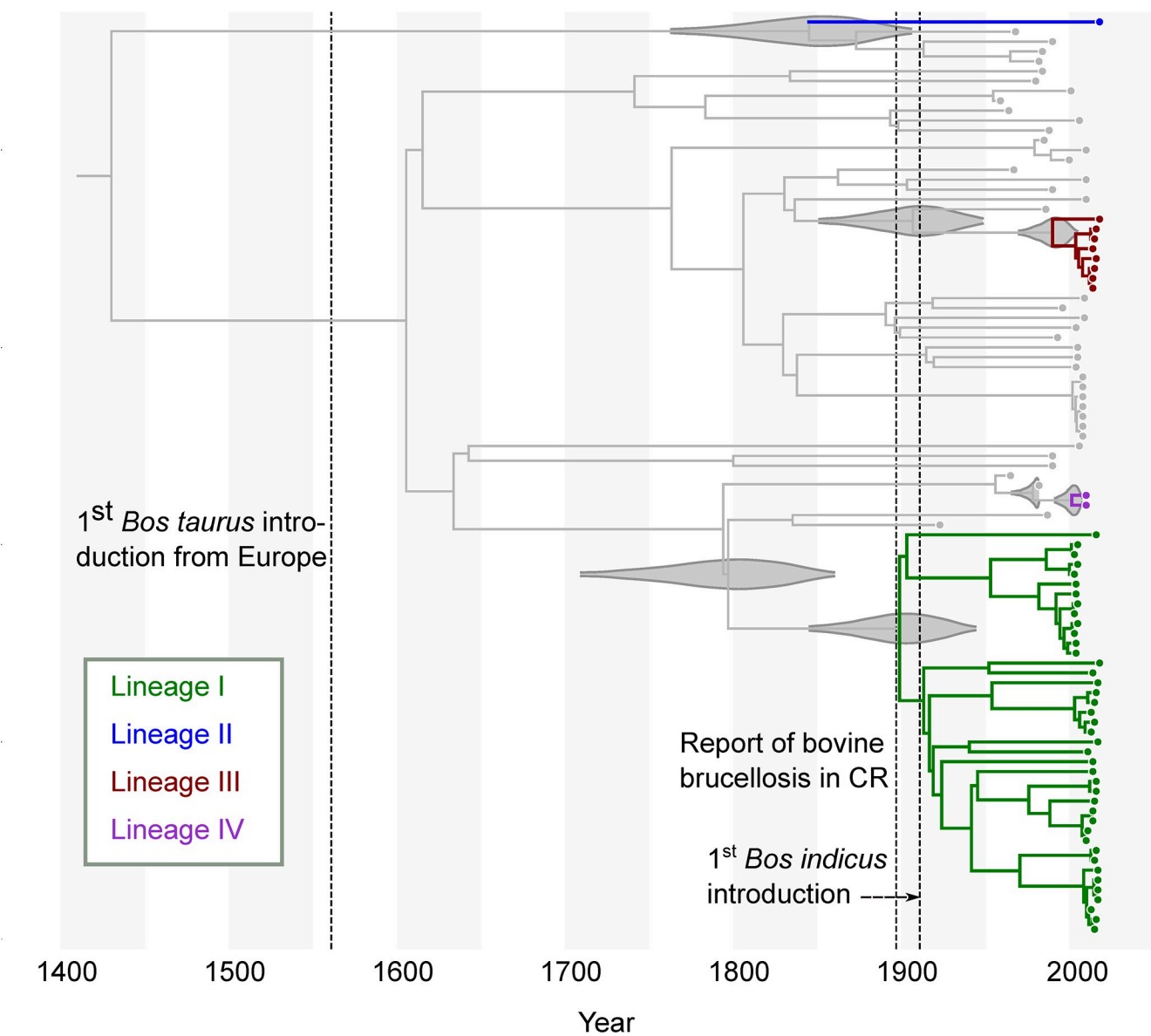

**Fig 4. Time-structured maximum clade credibility phylogenetic tree.** Molecular clock estimation based on an uncorrelated relaxed clock model with skyline tree prior. Colored branches represent the CR lineages. The 95% highest posterior probability distribution (HPD) for each upper and lower bound are shown as grey violin graphs on the respective nodes. Dotted lines illustrate three important events in livestock management in CR: (i) the first registered introduction of *Bos taurus* cattle from Europe in 1561, (ii) the introduction of bovine brucellosis during 1897–1902, and (iii) the first registered introduction of *Bos indicus* during 1911. Lineage V was excluded of the analysis because the year of isolation is unknown. Gray-white bars show a period of 50 years.

widely for genotyping studies worldwide [53,54]. In this study, the MLVA-16 results obtained did not match those obtained by phylogeny based on WGS. This difference may be explained by the higher resolution power provided by WGS and homoplasy of *B. abortus* VNTR markers, as previously proposed [55–58].

Our phylogenetic results support that the original *B. abortus* introductions came from South America (i.e., Colombia and Brazil) and North America (i.e. Mexico and the US) as it was previously suggested by MLVA-16 analysis [8]. This was not unexpected since these countries have been an important source of cattle to the region. The time-structured phylogenetic analysis indicated that the lineages were occurring contemporaneously: all genomes included

**Table 2. Highest posterior distribution interval and time (shown in AC years) to the most common recent ancestor (MRCA) from the five Costa Rican lineages.**

| Statistics | Lineage I | Lineage II | Lineage III | Lineage IV |
|---|---|---|---|---|
| **Upper bound (Ub) mean** | 1797 | 1845 | 1907 | 1978 |
| **Ub median** | 1796 | 1845 | 1907 | 1978 |
| **Ub 95% HPD interval** | 1709–1860 | 1763–1906 | 1851–1948 | 1966–1982 |
| **Lower bound (Lb) mean** | 1899 | NE[a] | 1990 | 2001 |
| **Lb median** | 1899 | NE | 1990 | 2001 |
| **Lb 95% HPD interval** | 1847–1944 | NE | 1970–2005 | 1991–2008 |

[a] NE: not estimated

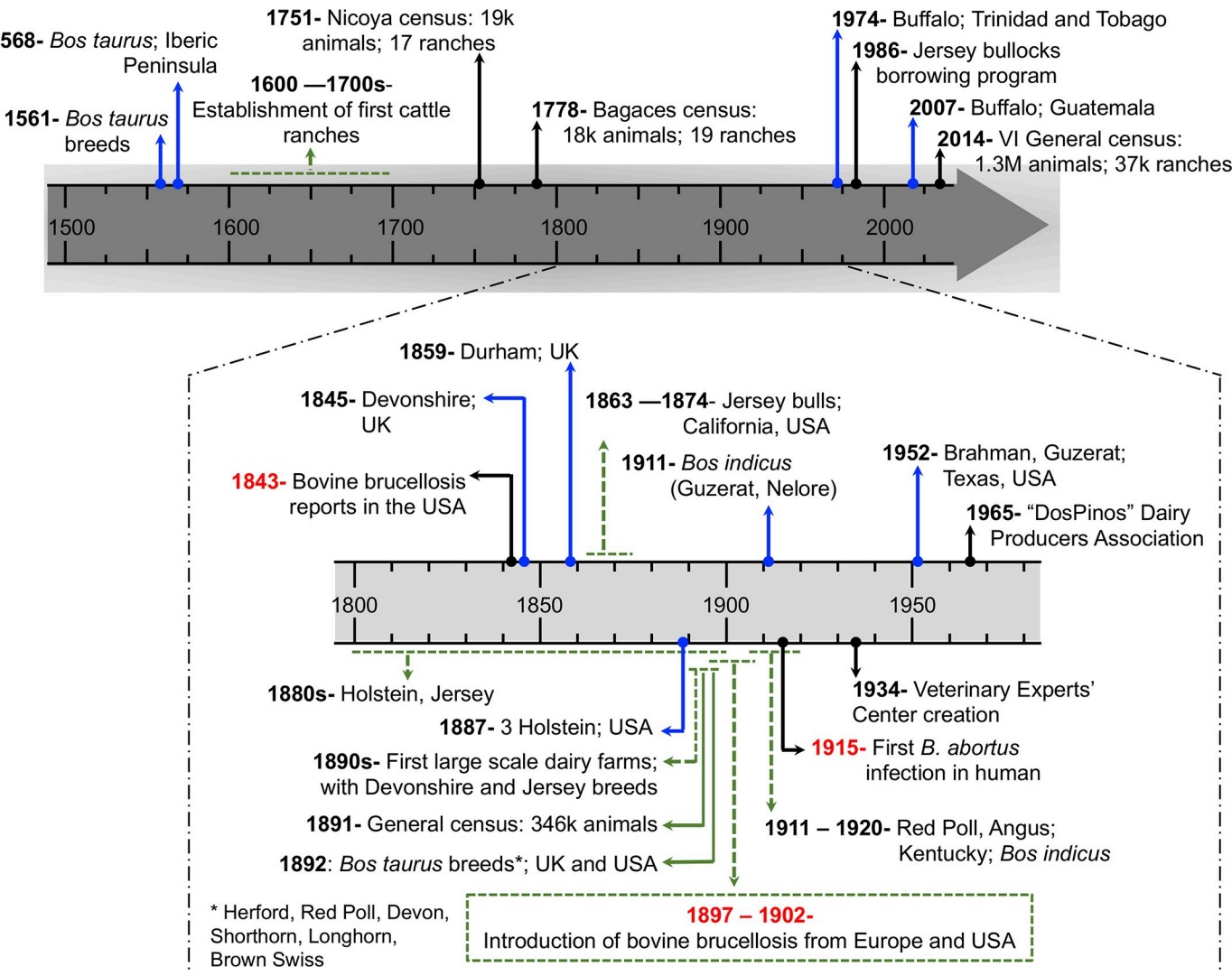

**Fig 5. Livestock management timeline in Costa Rica.** The main historical events in the agricultural development of the country are shown. Blue arrows point important introductions of cattle; when the specific geographical origin of the imported animals is known, it is indicated after a semicolon. Green arrows indicate events prolonged during a period of time. Black arrows mark important facts in cattle management. Red fonts highlight brucellosis related incidents. The period between 1800 and 1975 is expanded to increase resolution of an important number of changes occurred during that period in the livestock history of CR. The main cattle source for CR agricultural development were the US and UK, with some minor incursions from Central America and the Caribbean. The exponential livestock growth in CR is exemplified by the animal census, first restricted to small areas, and then compared to the last general census performed in 2014.

in the analysis were sampled from 2003 to 2018 and all of the lineages included samples from the last four years.

The 19[th] century was characterized by great ranching expansion in CR. Several *Bos taurus* breeds recognized for their good dairy performance, were introduced into the territory. During this time of development, bovine brucellosis was detected. The historical records identified this fact between 1897 and 1902 [13]. This period coincided with the temporal range shown by our phylodynamic analysis as the date for the MRCA of lineage I, the most widespread and abundant clade in CR. The movement of cattle across the country, potentiated by bull borrowing programs [13], could enable the dispersal of the new strains and could explain why most of the isolates clustered together.

The water buffalo isolate that comprised lineage II, clustered most closely with a 1997 Colombian isolate, but the other water buffalo isolates are phylogenetically closer to cattle isolates in lineages I and III. This suggests that at least three different lineages of *B. abortus* are infecting buffalo in CR. As we could not estimate the lower bound for lineage II, we can only infer that its upper bound MRCA may have occurred before introduction of water buffalo to T&T from India, during 1905 and 1908 [50].

Historical events, close to the date of the MRCA of lineage III, with median date 1990 (95% HPD: 1970–2004), are not so clear, but its proximity to isolates from the US points two possible sources: (i) the introduction of Red Poll and Angus breeds from Kentucky from 1911 to 1920, and (ii) the introduction of *Bos indicus* breeds from Texas in 1952 (Fig 5). Unfortunately, the metadata obtained from the hosts only specifies that the animals were of both breeds, not allowing us to determine which were *Bos indicus* and which *Bos taurus*. The youngest lineage IV, with a median estimated date at the year 2001, shows a more recent introduction, that could have originated from illegal introduction of semen and cattle from Nicaragua that have occurred sporadically from 1986 to 2011 [51].

Our estimated substitution rate was $8.28 \times 10^{-8}$ (95% HPD interval: $2.8 \times 10^{-8}$–$1.7 \times 10^{-7}$) per site per year using the HKY85 evolutionary model [59] and the discrete gamma model of heterogeneity among sites [60]. Here we estimated a lower substitution rate for *B. abortus* than Kamath et al. 2016 [61]. They reported $1.4 \times 10^{-7}$ substitutions per site per year (95% HPD interval: $1.09 \times 10^{-7}$–$1.73 \times 10^{-7}$); interestingly the 95% HPD of our estimated rate overlaps with their median value. Both results are similar to the $10^{-7}$ to $10^{-8}$ mutations per site per year of the intracellular microorganism *Mycobacterium tuberculosis* [62–64].

The variability among and within the lineages was represented by a small but evident number of SNP variations in the WGS of isolates from the outbreak in San Juan de Chicuá, comprised in the CR lineage I (Fig 1). This result should be judged according to temporal and spatial perspectives, as it is remarkable to find variability among *B. abortus* isolates obtained from the same host species in a restricted geographic area of only dairy farms. First, the isolates were obtained in a time span of three years. Second, the total area of San Juan de Chicuá is just 18.7 km$^2$, including at least eight dairy farms. Most of the SNPs were found at intergenic noncoding sequences positions; however, these SNP differences may influence the expression levels of neighbor CDS regions or RNA coding genes, probably contributing to niche adaptation or modifying bacterial virulence [65,66]. The variable positions of GIs, surrounded by mobile genes or insertion sequences [35], suggested genetic reorganization of regions that code for important CDS related to metabolism control and virulence. Further research with more isolates from different regions and hosts is required to define the implications of the GIs and IS*711* position differences.

The WGS analysis performed in this study reveals the presence of five lineages of *B. abortus* in CR and highlights that the phylodynamic calibration matches with the historical events related to the introduction of bovines into the country. Both gaps in sampling and under

sampling of some of the socioeconomic regions may have hampered the detection of more lineages in CR. This is relevant considering that the last strain detected as introduced into CR may have occurred as recently as 2007. From an epidemiological perspective, we do not know the relative virulence and infectious potential of the different *B. abortus* strains circulating in the country. Neither do we know if the different strains would have any influence on the vaccination and control programs. The presence of two lineages (lineages I and III) where there are genomes from isolates obtained from cattle, buffalo and humans, has not gone unnoticed and seems relevant. Transmission between these species is therefore a possibility in CR. In the case of lineage I, most of the human isolates differ in only 27 SNPs positions, regardless of the origin and time span. To further understand these observations in the context of animal and human infection, a One Health approach is required, with cross-sector collaboration of those involved in such tasks, including public health services, academia, medical and veterinary health agencies, private sector as well as regional and international organizations. Data related to animal movements both, within the country and coming to the country would be of great value in order to track lineages spread and the introduction of new ones.

The permanence of the originally introduced lineages highlights the unsuccessful outcome of the disease control mechanisms thus far implemented in CR. The presence of recent lineages revealed gaps in the control of animal movement and brucellosis surveillance. Following this, our results highlight the importance of the WGS metadata of microbial samples for keeping track and understanding the epidemiology of relevant infectious diseases and propose the use of phylodynamic analysis as a model to study the introductions of brucellosis into new regions.

## Supporting information

**S1 Dataset. MLVA-16 and WGS metadata.** MLVA profiles and metadata of *B. abortus* isolates included in the study.
(XLSX)

**S2 Dataset. Maximum likelihood estimations (MLEs) for molecular clock models and population structure analysis.** MLEs for molecular clock models by path sampling, stepping-stone sampling and Bayes factor for each one of the tested clock models and tree priors are shown. Partition and MLEs for the population structure using RhierBAPS are shown.
(XLSX)

**S3 Dataset. SNPs information.** Details of SNPs in the alignment, the reference and its meaning are shown individually for each one of the Costa Rican genomes used in the study. Specific SNPs per lineage are presented in separate spreadsheets.
(XLSX)

**S1 Fig. ACT comparison of 23 anomalous regions/genomic islands (GIs) in representative *B. abortus* genomes, selected from the CR lineages.** Islands were concatenated and ordered in a pseudo-molecule; this is represented as the upper gray-black blocks. Top coordinates show relative size in bp of each island. Dotted gray line represents the division of both chromosomes. Each comparison box shows the islands' distribution in the query genomes. Regions present in the same position and order in the genomes (when compared to the pseudomolecule) are shown in red color, and inversions in blue. Absence of segments of the islands are shown as white spaces in the boxes. Independently of the presence of inversions, different rearrangements of the genes included in the islands are observed among the isolates. CR lineages are indicated by the colors of the branches and at the right side of the boxes: lineage I, green;

lineage II, blue; lineage III, maroon; lineage IV, purple; lineage V, orange.
(PDF)

**S2 Fig. IS*711* and Tn*2020* insertion signature patterns for *B. abortus*.** Each peak represents the location of at least 50X coverage, 99% identity IS*711* insertion (peak in black color) and Tn*2020* (peak in red color). The position in the first and second chromosomes (shown as a concatenated molecule) is indicated by the scale bar (in Mb) above. The smaller maroon bars next to the tips indicate the representative genomes used for the analysis. CR lineages are indicated by a colored bar next to the tips.
(PDF)

## Acknowledgments

The authors are grateful to Daphnne Garita, Eunice Víquez, Andrés Balbin, and Reinaldo Pereira for their technical assistance, INCIENSA Costa Rica, Kate S. Baker and Gordon Dougan for helpful discussions.

## Author Contributions

**Conceptualization:** Jason T. Ladner, Jeffrey T. Foster, Nicholas R. Thomson, Edgardo Moreno, Caterina Guzmán-Verri.

**Data curation:** Marcela Suárez-Esquivel, Gabriela Hernández-Mora, Nazareth Ruiz-Villalobos, Elías Barquero-Calvo, Carlos Chacón-Díaz, Jason T. Ladner, Norman Rojas-Campos, Caterina Guzmán-Verri.

**Formal analysis:** Marcela Suárez-Esquivel, Gabriela Hernández-Mora, Nazareth Ruiz-Villalobos, Elías Barquero-Calvo, Jason T. Ladner, Jeffrey T. Foster, Esteban Chaves-Olarte, Edgardo Moreno, Caterina Guzmán-Verri.

**Funding acquisition:** Elías Barquero-Calvo, Esteban Chaves-Olarte, Nicholas R. Thomson, Edgardo Moreno, Caterina Guzmán-Verri.

**Investigation:** Marcela Suárez-Esquivel, Nazareth Ruiz-Villalobos, Gerardo Oviedo-Sánchez, Norman Rojas-Campos, Caterina Guzmán-Verri.

**Project administration:** Caterina Guzmán-Verri.

**Resources:** Gabriela Hernández-Mora, Elías Barquero-Calvo, Carlos Chacón-Díaz, Jeffrey T. Foster, Norman Rojas-Campos, Esteban Chaves-Olarte, Nicholas R. Thomson, Edgardo Moreno, Caterina Guzmán-Verri.

**Supervision:** Caterina Guzmán-Verri.

**Validation:** Gabriela Hernández-Mora, Elías Barquero-Calvo, Carlos Chacón-Díaz, Jason T. Ladner, Jeffrey T. Foster, Caterina Guzmán-Verri.

**Visualization:** Marcela Suárez-Esquivel, Nazareth Ruiz-Villalobos, Jason T. Ladner, Gerardo Oviedo-Sánchez, Jeffrey T. Foster.

**Writing – original draft:** Marcela Suárez-Esquivel.

**Writing – review & editing:** Marcela Suárez-Esquivel, Gabriela Hernández-Mora, Nazareth Ruiz-Villalobos, Elías Barquero-Calvo, Carlos Chacón-Díaz, Jason T. Ladner, Gerardo Oviedo-Sánchez, Jeffrey T. Foster, Norman Rojas-Campos, Esteban Chaves-Olarte, Nicholas R. Thomson, Edgardo Moreno, Caterina Guzmán-Verri.

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
