## [Decision Letter · Decision Letter 0]

11 Dec 2019

Dear Guzman-Verri:

Thank you very much for submitting your manuscript "Persistence of Brucella abortus lineages revealed by genomic characterization and phylodynamic analysis" (#PNTD-D-19-01458) for review by PLOS Neglected Tropical Diseases. Your manuscript was fully evaluated at the editorial level and by independent peer reviewers. The reviewers appreciated the attention to an important problem, but raised some substantial concerns about the manuscript as it currently stands. These issues must be addressed before we would be willing to consider a revised version of your study. We cannot, of course, promise publication at that time.

We therefore ask you to modify the manuscript according to the review recommendations before we can consider your manuscript for acceptance. Your revisions should address the specific points made by each reviewer. 

When you are ready to resubmit, please be prepared to upload the following:

(1) A letter containing a detailed list of your responses to the review comments and a description of the changes you have made in the manuscript.

(2) Two versions of the manuscript: one with either highlights or tracked changes denoting where the text has been changed (uploaded as a "Revised Article with Changes Highlighted" file); the other a clean version (uploaded as the article file).

(3) If available, a striking still image (a new image if one is available or an existing one from within your manuscript). If your manuscript is accepted for publication, this image may be featured on our website. Images should ideally be high resolution, eye-catching, single panel images; where one is available, please use 'add file' at the time of resubmission and select 'striking image' as the file type. 

Please provide a short caption, including credits, uploaded as a separate "Other" file. If your image is from someone other than yourself, please ensure that the artist has read and agreed to the terms and conditions of the Creative Commons Attribution License at http://journals.plos.org/plosntds/s/content-license (NOTE: we cannot publish copyrighted images). 

(4) If applicable, we encourage you to add a list of accession numbers/ID numbers for genes and proteins mentioned in the text (these should be listed as a paragraph at the end of the manuscript). You can supply accession numbers for any database, so long as the database is publicly accessible and stable. Examples include LocusLink and SwissProt.

(5) To enhance the reproducibility of your results, we recommend that you deposit your laboratory protocols in protocols.io, where a protocol can be assigned its own identifier (DOI) such that it can be cited independently in the future. For instructions see http://journals.plos.org/plosntds/s/submission-guidelines#loc-methods

While revising your submission, please upload your figure files to the Preflight Analysis and Conversion Engine (PACE) digital diagnostic tool, https://pacev2.apexcovantage.com/ PACE helps ensure that figures meet PLOS requirements. To use PACE, you must first register as a user. Then, login and navigate to the UPLOAD tab, where you will find detailed instructions on how to use the tool. If you encounter any issues or have any questions when using PACE, please email us at figures@plos.org.

We hope to receive your revised manuscript by February 11, 2020. If you anticipate any delay in its return, we ask that you let us know the expected resubmission date by replying to this email.

To submit a revision, go to https://www.editorialmanager.com/pntd/ and log in as an Author. You will see a menu item call Submission Needing Revision. You will find your submission record there. 

Sincerely,

Elaine Maria Seles Dorneles, Ph.D.

Guest Editor

Ana LTO Nascimento

Deputy Editor

Reviewer's Responses to Questions

**Key Review Criteria Required for Acceptance?**

**Methods**

-Are the objectives of the study clearly articulated with a clear testable hypothesis stated?

-Is the study design appropriate to address the stated objectives?

-Is the population clearly described and appropriate for the hypothesis being tested?

-Is the sample size sufficient to ensure adequate power to address the hypothesis being tested?

-Were correct statistical analysis used to support conclusions?

-Are there concerns about ethical or regulatory requirements being met?

Reviewer #1: Yes

Reviewer #2: study design appropriate to address the stated objectives。

Reviewer #3: The objectives of the study clearly articulated with a clear testable hypothesis stated. The description of the isolates was performed in more detail in the results section. Information on host and location of the isolates should be in the methodology section. In the results section, the most important to be presented is not the characterization of the isolates studied as to their basic information, but the outcome of the proposed analyzes.

**Results**

-Does the analysis presented match the analysis plan?

-Are the results clearly and completely presented?

-Are the figures (Tables, Images) of sufficient quality for clarity?

Reviewer #1: Yes

Reviewer #2: analysis data were match the analysis plan, and results was clearly and completely presented.

Reviewer #3: The results are well presented, however, it is recommended to review Figure 3: Please check the color of the subtitle in Clades 1 and 2, it seems to me that Clade 1 should be represented in green, and Clade 2 in blue, following the logic of the previous figures and also due to the amount of isolates in each one.

**Conclusions**

-Are the conclusions supported by the data presented?

-Are the limitations of analysis clearly described?

-Do the authors discuss how these data can be helpful to advance our understanding of the topic under study?

-Is public health relevance addressed?

Reviewer #1: Yes

Reviewer #2: The conclusions supported by the data presented, but sepecies had no observed in conlusion.

Reviewer #3: The findings of the study are in line with the objective and demonstrated the importance and applicability of the results in promoting a better understanding of animal brucellosis epidemiology. Although no results related to human isolates have been discussed, which was acknowledged by the authors, the study makes an interesting contribution to public health. In the context of one health it is essential to recognize the importance of a good understanding of the dynamics of infections in animals in order to prevent disease in human population.

**Editorial and Data Presentation Modifications?**

Reviewer #1: (No Response)

Reviewer #2: substantial revision

Reviewer #3: Line 118: It is not clear in S1 which 188 isolates were analyzed in this study.

Line 120: Specify how many samples were obtained through the brucellosis national surveillance programs or national hospitals.

Line 149: Explain here the reason for sequencing this isolate with a different technique (Line 273 to 276).

Line 154: In supplementary material S1, when using the "This study" filter only 53 records are found. Incompatible or unintuitive to find which are the 95 genomes used in this analysis. Please check and correct.

Line 177: Quote abbreviation for CoDing Sequence (CDS) in the first time it appears in text.

Line 278: As a suggestion, it would be interesting for the circles to have a size proportional to the number of isolates they represent.

Line 279: Lineage II is represented with a blue sign only in the province of Chorotega. Lineage III in turn seems to be more widely distributed. Please check.

Line 297 and 298: Phrase more appropriate in discussion section than in presentation of results.

Paragraphs of lines 309 to 316: Avoid excessive use of the pronoun "we".

Line 331 to 334: Phrase more appropriate in discussion section than in presentation of results. Attention, please insert reference to Kamath et al., 2016.

Line 373 to 376: Phrase more appropriate in discussion section than in presentation of results.

**Summary and General Comments**

Reviewer #1: (No Response)

Reviewer #2: (No Response)

Reviewer #3: Although there is no clear explanation for the results observed for human isolates, it is important that the authors at least hypothesize what was observed. The results should at least provide guidance and suggest possible strategies for further investigations in the epidemiology of human brucellosis.

PLOS authors have the option to publish the peer review history of their article (what does this mean?). If published, this will include your full peer review and any attached files.

Reviewer #1: No

Reviewer #2: No

Reviewer #3: No

---

## [Decision Letter · Decision Letter 1]

19 Feb 2020

Dear Dr. Guzman-Verri,

Thank you very much for submitting your manuscript "Persistence of Brucella abortus lineages revealed by genomic characterization and phylodynamic analysis" for consideration at PLOS Neglected Tropical Diseases. As with all papers reviewed by the journal, your manuscript was reviewed by members of the editorial board and by several independent reviewers. The reviewers appreciated the attention to an important topic. Based on the reviews, we are likely to accept this manuscript for publication, providing that you modify the manuscript according to the review recommendations. 

Sincerely,

Elaine Maria Seles Dorneles, Ph.D.

Guest Editor

Ana LTO Nascimento

Deputy Editor

Reviewer's Responses to Questions

**Key Review Criteria Required for Acceptance?**

**Methods**

-Are the objectives of the study clearly articulated with a clear testable hypothesis stated?

-Is the study design appropriate to address the stated objectives?

-Is the population clearly described and appropriate for the hypothesis being tested?

-Is the sample size sufficient to ensure adequate power to address the hypothesis being tested?

-Were correct statistical analysis used to support conclusions?

-Are there concerns about ethical or regulatory requirements being met?

Reviewer #2: (No Response)

Reviewer #3: Despite finding corrections in each comment at the end of the file, the attached manuscript is the same as the first one previously submitted, and not the corrected version. Please check and resend.

**Results**

-Does the analysis presented match the analysis plan?

-Are the results clearly and completely presented?

-Are the figures (Tables, Images) of sufficient quality for clarity?

Reviewer #2: (No Response)

Reviewer #3: Despite finding corrections in each comment at the end of the file, the attached manuscript is the same as the first one previously submitted, and not the corrected version. Please check and resend.

**Conclusions**

-Are the conclusions supported by the data presented?

-Are the limitations of analysis clearly described?

-Do the authors discuss how these data can be helpful to advance our understanding of the topic under study?

-Is public health relevance addressed?

Reviewer #2: (No Response)

Reviewer #3: Despite finding corrections in each comment at the end of the file, the attached manuscript is the same as the first one previously submitted, and not the corrected version. Please check and resend.

**Editorial and Data Presentation Modifications?**

Reviewer #2: (No Response)

Reviewer #3: Despite finding corrections in each comment at the end of the file, the attached manuscript is the same as the first one previously submitted, and not the corrected version. Please check and resend.

**Summary and General Comments**

Reviewer #2: (No Response)

Reviewer #3: Despite finding corrections in each comment at the end of the file, the attached manuscript is the same as the first one previously submitted, and not the corrected version. Please check and resend.

PLOS authors have the option to publish the peer review history of their article (what does this mean?). If published, this will include your full peer review and any attached files.

Reviewer #2: Yes: zhenjun li

Reviewer #3: No
---

## [Editor Report · Decision Letter 2]

18 Mar 2020

Dear Dr. Guzman-Verri,

We are pleased to inform you that your manuscript 'Persistence of Brucella abortus lineages revealed by genomic characterization and phylodynamic analysis' has been provisionally accepted for publication in PLOS Neglected Tropical Diseases.

Best regards,

Elaine Maria Seles Dorneles, Ph.D.

Associate Editor

Ana LTO Nascimento

Deputy Editor

---

## [Editor Report · Acceptance letter]

8 Apr 2020

Dear Dr. Guzman-Verri,

We are delighted to inform you that your manuscript, "Persistence of Brucella abortus lineages revealed by genomic characterization and phylodynamic analysis," has been formally accepted for publication in PLOS Neglected Tropical Diseases.

Best regards,

Serap Aksoy

Editor-in-Chief

Shaden Kamhawi

Editor-in-Chief
